# The Promises and Limitations of Educational Tiers for Special and Inclusive Education

James M. Kauffman 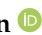

Department of Special Education, University of Virginia, Charlottesville, VA 22904, USA; jmk9t@virginia.edu

**Abstract:** Making public school accommodating of all learners such that the need for special education is obviated, or at least reduced, has long been a desideratum of educators. Various strategies for making general public education more accommodating of students with disabilities have been tried. The most recent efforts to improve the general education of students with disabilities involve various models of tiered education. Educational tiers can be logical and advantageous in some ways, holding promise for improving general education, but they do not address the core problems of special education. Special education is still needed as part of inclusive education.

**Keywords:** tiers; levels; teaching disabilities

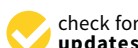



The idea that special education can and should become an educational invention of the past has current proponents, e.g., [1–4]. However, that idea is at least a half-century old. In 1970, special educator Evelyn Deno suggested that special education might be able to work itself out of business by teaching general educators to do special educators' jobs [5]. In later decades of the 20th century, others suggested merging special and general education or making general education so "supple" that it could serve all students and special education would no longer be needed [6–9]. Longing for an end to what special education is and what it does has a tortuous history [10–12].

With the concept of multiple tiers of support in general education, it appears some would argue that general education can "go it alone" or fully integrate the special into the general education so that a single, fully unified or integrated, inclusionary education system can be achieved, providing equity for all learners. For example, the SWIFT Schools eb site's home page [3] includes these statements: "We believe together we can transform education so that it benefits each and every student . . . " and "Leading the nation in equity-based Multi-tiered System of Support and inclusive education research and services."

As mentioned, the basic idea of merging special and general education is decades old, e.g., [6,13,14], but its current iteration is known as tiered education or multi (more than two) tiers, typically known as multi-tiered systems of support (MTSS). In some current plans for MTSS, e.g., [15], special education still exists, and lower tiers appear to be preliminary interventions that might prevent referral for special education. However, at least three things are not clear in all cases: (1) the tier that is designated as special education; (2) U.S. law (Individuals with Disabilities Education Act, IDEA) or other legal protections granted with each tier other than the one designated as special education; and (3) the qualifications other than a general education teaching license that are required to teach tiers other than the first.

## 1. Specialization in Education

Education is among the endeavors many consider so simple that specialization is not required in teaching basic skills [16]. Teaching such things as art, music, and physical education may be required at all levels of education, and specialization in teaching in particular curricular areas such as mathematics, science, history, etc., may be required at the secondary level. However, some seem to argue that no *special* education or separate

degree program is needed for teaching students with disabilities, only improved general education that includes some instruction in meeting the needs of *all* students [3,4].

In this respect, those who propose full inclusion—*only* inclusive education—must assume that teaching is much like ditch-digging. If you can dig a ditch, it matters little whether you are digging a trench for a sewer line or a water line. Ditch-digging requires no special training depending on what is to be put in the ditch. If you can dig a ditch for one thing, then you can dig a ditch for anything. Education stands very much alone as an endeavor (or profession) in which specialization for students with disabilities is thought to be—even argued to be—unneeded [1–4]. The assumption seems to be that if you can teach one child to do something, then you can teach any child to do it. A common disparagement of teaching any group is the comment that teaching is not rocket science. In fact, it is more complicated than rocket science. Additionally, it becomes more complicated with increases in the size and diversity in prior learning and cognitive ability of the group to be taught. One group of scholars wrote:

> Teachers who take their task seriously understand the ignorance of someone who asks, "Who knew teaching could be so complicated?" Experienced, competent teachers also understand how adding to the learning diversity of a group of students (not the group's racial, ethnic, gender, or other diversities that do not determine learning) adds to the difficulty of effective instruction. As with virtually any task, some will claim that whatever activity (teaching, building, playing a musical instrument or sport, etc.) is easy—or claim to have a simple solution to the challenge of its mastery. For more than 45 years, some special education leaders have supported the fiction that general educators should be able, at least with help from special educators at their elbows, to teach all children without exception, including those with disabilities . . . .

> In education, differentiation is often presented as an easy, or at least eminently doable, solution to teaching diverse groups. Inclusion of the most difficult students in general education is sometimes presented as something all teachers worth their salt can accomplish with a little extra effort, a little help, and/or reasonable determination. Aspersions are then cast on good general education teachers who say they cannot do it or cannot do it well. We hope that one legacy of the inclusion movement in education will be better understanding of the complexities and demandingness of teaching. [16] (pp. 261–262)

Most people readily recognize the absurdity of propositions such as the following: (a) *all* drivers will be licensed to drive *all* vehicles, with no special training or licensure to drive any trucks, buses, heavy equipment, or other vehicles not airborne; (b) *all* pilots will be expected to fly *all* airplanes for *all* purposes, regardless of number or type of engine(s), size, or purpose; (c) *all* builders will be licensed to construct all types of buildings; (d) *all* physicians will be licensed to perform all medical treatments, including examinations, prescriptions, surgeries, and other medical procedures; (e) *all* hospitals will be open to *all* patients, and *all* patients will be placed in general medical units regardless of medical condition or diagnosis; (f) *all* soldiers will be expected to operate *all* weapons of defense and be trained to accomplish *all* missions; (g) *all* lawyers will be expected to handle *all* cases involving law, regardless of the nature of the case; (h) *all* teachers will be prepared to teach *all* subjects at *all* levels; and (i) *all* dentists will provide *all* dental services and procedures, including extractions, orthodontia, and dental implants.

However, the insistence that appropriate education for *all* students can be achieved in general education—one system so "flexible" or "supple" that no special education is needed, that no student needs to be "singled out," "labeled," or "segregated" from general education is puzzling. Presumably, when "all" is used, and especially when the phrase "all means all" is invoked, the reference is to each and every individual, no exceptions. Perhaps, those using "all" and saying "all means all" do not really mean *all* in a *literal*



sense. However, it is incumbent upon them to say so and to describe the exceptions in some manner—to state the criteria or process for making exceptions.

Knowing that the number of exceptions needed to disprove the claim that "all means all" is precisely *one* and that certainly more than one living child will not be appropriately served by "inclusive" general education, some nevertheless double down on the claim e.g., [3,4]. Observations of inequities based on diversities other than disability (e.g., color, ancestry, gender) have been used to justify having only "inclusive" education, with special education tagged as "exclusionary," "segregated," and "othering" e.g., [4]. What is lost in the justifiable objections to inequities based on other forms of diversity is the nature of differences—the fact that disability is a different *kind* of diversity demanding a different response from educators concerned about equity.

## 2. Attempts to Make Education Appropriate for All

Many attempts have been made to find a method, structure, or ideology that makes the promise of appropriate education for all a reality without having special education, to make schools inclusive of all students without identifying or "separating out" any for education away from the general group.

Grades, levels, classes, and subjects (i.e., curriculum areas) are obviously ways of "separating out" students for particular instructional activities, but these are not usually considered "segregated" groupings, whereas any separate special education is called "segregated." Special education bears the brunt of condemnation for sorting, labeling, and segregating. However, all programmatic groupings in education require sorting, labeling, and segregating to meet their objectives. As with special education, they are dedicated to a particular activity and purpose for particular students. Special education, too, is better described as "dedicated" than "segregated" [17]. The moral taint of segregation is unnecessarily and unjustly attributed to special education.

Among the attempts to make general education more accommodating of students with disabilities is the idea of "pre-referral teams," groups of teachers (general and special, and perhaps school psychologists or counselors) who try to problem-solve the education of a particular student to preclude or prevent referral of that student for special education evaluation. The assumption of pre-referral is that the general education teacher has not tried techniques or strategies that would resolve the problem(s) that could lead to referral of a student about whom the teacher is concerned for the evaluation for special education.

Attempts to improve general education's responses to students with disabilities also included the regular education initiative of the 1980s (REI, peculiar to the United States) [14] and response to intervention or instruction of the 1990s (RTI; perhaps invented in the United States but not only applicable to teaching practices there) [18–22]. Although RTI has been suggested as a way of making full inclusion (i.e., placement of *all* students with disabilities in general education) possible, a more recent and internationally lauded idea about how this might be accomplished is the notion of tiers within general education. MTSS is usually focused on academic issues, and another tiered system called PBIS (positive behavioral interventions and supports) is usually focused on behavioral issues.

## 3. The Development of Tiered Models

The ideas leading to tiers have a history beginning in the late 20th century. In the early 21st century, many different models with a variety of acronyms have been developed [23]. The basic concerns leading to the invention of tiers include:

1. Many general education teachers do not use evidence-based instructional and behavior management practices, leading to unnecessary academic and behavioral problems;
2. Many students need help in improving their academic learning and emotional/social behavior but do not have actual disabilities;
3. Students' problems often become severe because intervention is delayed too long and opportunities for prevention are overlooked;

4.      Students are often mistakenly identified as having disabilities because of these three previously stated concerns;
5.      Too many students are served by special education simply because that is the only special service they can obtain in schools.

Consequently, Tier 1 consists of good evidence-based instructional and behavior management practices, a primary prevention strategy; Tier 2 is an attempt to catch problems early, using interventions known to be effective in secondary prevention so that students' problems do not become severe; and Tier 3 is intensive, individualized, targeted, interventions associated with tertiary prevention, managing problems so that they do not become overwhelming.

One thing not clear is precisely how the implementation of tiers is not a form of "tracking." In one sense, it appears to be a refined form of tracking, in that observed differences in students' learning and behavior are used to justify different designations and instruction. Perhaps it is more explicit, defensible, flexible, but relabeled tracking.

Many other issues involving tiers in the general case have been noted [22]. The most sophisticated tiered model to date combines academic, behavioral, and social concerns into a comprehensive, integrated, three-tiered model (Ci3T) [18]. In the Ci3T model, special education still exists but is independent of tiers. Specifically, students identified as having a disability are not necessarily assigned to Tier 3 but may be found in any tier, depending on their IEP (individual education program, required by U.S. law for all students receiving special education). Ci3T does have advantageous features, including improved general education and the possible inclusion of more students at all levels of general education [15]. Nevertheless, recognition of what any program of tiers greater than two (general/special) can and cannot do is important.

## 4. What More Than Two Tiers Can and Cannot Do

Although some may believe that tiers of general education can result in the inclusion of *all* students with disabilities in typical classrooms in neighborhood schools [3,4], it is important to note that not all proponents of tiers are advocates of full inclusion. However, some advocates of tiers argue that labeling and stigma can be avoided. They might even suggest that a child is not actually "in" a tier but receiving the supports offered in that tier. Nevertheless, someone must decide of students, "*This* one, not *that* one" will receive the programs or supports of a given tier, i.e., individuals must be chosen to receive the services of a given tier. Someone must decide which students will receive which services. Classification, sorting, labeling—actually doing something to address the diversity of responses to teaching—are things that cannot be avoided unless everyone is to receive the same thing. Furthermore, a student receiving the supports of Tier 1, for example, will inevitably be called a Tier 1 student. Moreover, any tier greater than the first will inevitably be stigmatizing, i.e., all tiers higher than Tier 1 will be stigmatizing, and higher tiers with higher numbers will carry more stigma than those with lower numbers.

Having more than two tiers (the traditional special/general education framework) so that there is a "sort of" or quasi-special education to address problems that are not considered actual disabilities may be a very good idea. In fact, tiers hold promise as a way of improving general education. However, it is also important to ask the following of any proposed model of tiers:

1.      Precisely what qualifies a student to receive services or interventions associated with each tier?
2.      What legal protections and regulations apply to each tier?
3.      What preparation or qualifications are needed to implement the procedures of each tier?
4.      How are tiers related to special education?

A given tier might or might not be designated as special education. Indeed, some have stated that no tier is exclusive to special education, and that special education identification

and IEPs are independent of tiers [15]. However, this suggests additional questions. For example:

1.  Just how is a student identified initially for special education—by what measure of achievement or need?
2.  Must a child first be found to need the most intensive, individualized interventions associated with the highest tier before being found eligible for special education?

Important to recognize is what tiers can and cannot do. One thing they *can* do is add to teachers' options for levels or types of instruction and behavior management. In that respect, they hold considerable promise for improving general education. However, at least in the area of managing behavior, Tier 2 interventions seem to require individualization because no behavioral intervention works reliably with all students [24,25]. If, in fact, this is the case, and particularly if individualized attention or programs are found to be necessary for success with Tier 1, then the value of designating tiers might be questioned.

The things any model of tiers *cannot* do are:

1.  Avoid labels;
2.  Avoid "*This* one, not *that* one" decisions;
3.  Avoid the stigma associated with all tiers greater than the lowest;
4.  Avoid either labeling one tier as special education and granting all the legal regulations and protections associated with that tier—if those regulations and protections are to be maintained—or specifying just how students are to be identified as having a disability, if they are disabled;
5.  Avoid the issue of legal regulations and protections that should accompany tiers greater than one but less than the highest.

Inclusion is an important aspect of education for students with disabilities, but so is having special education in environments other than general education [24,26]. In fact, insistence on the inclusion and elimination of special education might be predictably self-defeating [26–30]. Levels or tiers of education, regardless of the number of them or their comprehensiveness and integration, do not address the three core problems of special education—the two-tiered system of education (general/special). These three core problems are:

1.  Drawing a line that separates special education from general education, one that is chosen from continuous distributions of academic performance and problematic behavior [31];
2.  Deciding just where (or in what environment) a particular student should be taught, chosen from a continuum of alternative placements [32];
3.  Prescribing precisely how and by whom particular students should be taught [33].

## 5. Concluding Remarks

Tiers are a good concept in many ways, and having more than two of them could allow more mainstreaming or inclusion of students in general education. Nevertheless, more tiers do not address, much less solve, the three core problems of special education noted in the preceding paragraph. The danger is not only that multiple tiers will prove to be another "latest thing" or "fad" of education [34], but that they will be used as an opportunity to dismantle special education, becoming a "new normal" [10], and will become a "size" that, presumably, fits all [26].

Attractive ideas can and have been used to mislead educators and the public into policies that are found to be unworkable [29,35]. To date, efforts to reform American public education in significant ways have failed. Tiers are an attractive idea, but so far little or no direct evidence of the use of tiers has demonstrated that tiers prevent problems or reduce the need for special education. This is not to say that no components of tiers that are practiced well have shown no promise in the prevention or reduction in problems or need for special education. Indeed, some have.

Teaching most students well is not easy. Teaching those with disabilities and doing it well is particularly challenging. Inclusive school environments have proven to be less successful than specialized instructions in separate settings for the most difficult to teach (i.e., lowest-performing) students [36]. Implementing the best models of tiered education with great fidelity is not easy, even in experimental situations, and bringing the implementation of them to scale such that they are practiced reliably in most schools would be a Herculean task, one probably quite unlikely to be accomplished. More likely is that many schools will claim to be implementing tiers but make that claim as a reason for eliminating special education or important parts of it (e.g., identifying disabilities that are not immediately obvious to almost all observers).

Some who have studied alternative numbers of tiers have found making Tier 1 work well to be very difficult. Some good advice from them: keep things as simple as possible, and note that teachers have limitations.

> By suggesting that some schools consider implementing a two-tier rather than three-tier framework, we are not saying that less complex frameworks are as effective as more complex ones. In principle, we would expect a three-tier approach to be more successful with more children. But in reality, many schools are *not* deciding between three and two tiers. They are struggling just to make Tier 1 work. . . . In considering a two-tier alternative, it is important to remember that the conventional three-tier approach, however logical and consistent with best practices, is without empirical validation. We do not say this dismissively or to be contentious or provocative. Rather, our point is simply that there is nothing sacrosanct about a three-tier framework. [37] (p. 266)

Of course, it is possible to argue for any number of tiers, including zero. The notion of having a carefully planned, individualized program of education for all students (essentially, having as many tiers as there are students) is, of course, risible. Some seem to argue for no tiers at all, simply doing away with the idea of having special education or, as it is known in some nations, SEN (special educational needs) [1–4,38]. The idea is that identifying disabilities or special educational needs is "othering" that justifies exclusion and segregation [4].

Education having no tiers at all, however, and denying the need of some students for *special* education, is at root denial not only of science generally and a science of education in particular, but also denial of the very humanity of students with disabilities and of teachers. It does recognize the fact that students with disabilities share many or most of the basic human needs of *all* students, but it also denies their human need for *education* which is not at all like the *educational* needs of most other students [39]. It denies the very human need and right of *all* students for *appropriate* education, plus the need of teachers for work they can do well. It makes education nothing different from any other civic activity, treating it as if being appropriately educated means merely being in a specified classroom.

The best number of tiers greater than zero is debatable, but having zero tiers is a profoundly regressive notion. It takes education back to the days before any *special* needs were recognized, and students with disabilities were expected to sink or swim like everyone else in the mainstream. It is a refusal to recognize the extraordinary in education for what it is.

The United Nations Convention on the Rights of Persons with Disabilities (CRPD) addresses the issue of education, although not including the quality or specialized nature of education for students with disabilities [39]. It addresses inclusion as a matter of a right to place only, and does not address the many practical issues of making sure such children receive appropriate education. It does not address the issue of tiers, apparently assuming that no tiers at all are necessary. Unfortunately, while addressing many important rights of persons with disabilities, it confuses " . . . the stone of 'being there' and the bread of learning critical skills . . . " [40] (p. xi). This is nothing short of a moral catastrophe.

**Funding:** This research received no external funding.

**Institutional Review Board Statement:** Not applicable.

**Informed Consent Statement:** Not applicable.

**Data Availability Statement:** Not applicable.

**Conflicts of Interest:** The author declares no conflict of interest.

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
