# Peer review of "The Promises and Limitations of Educational Tiers for Special and Inclusive Education"

_education, doi:10.3390/educsci11070323_

Round 1

Reviewer 1 Report

I feel that further engagement with the purposes of, ideologies behind, and ramifications of educational tiers/ multi-tiered systems of support (MTSS) a worthy subject to be interrogated. As I suspect the authors would agree (and in some cases they stated) multi-tiered systems of supports have been widely implemented, and often without strong fidelity in practice or clear rationale between the lines of special education and tiered interventions. This laudable goal, though, is not achieved in this particular paper in a way that clearly understands the purposes of MTSS nor the goals of inclusionists or the boundaries of general and special education. 

The paper itself makes a lot of assumptions that are not based in current literature, the current direction of the field(s) of MTSS and inclusive education, particularly in connection to general education.

Also, as an overarching concern, the abstract claims that the authors are responding to a "danger of tiered education is that it will be used as an excuse to eliminate special education." However I don't see this argument really being made in the paper itself

I will start at the top of the paper to break down some more specific concerns:

  1. Line 26: the fear claimed is that special educators are proposing to do general educators jobs for them. I can see the authors argue that these knowledge-bases and systems should be separate fields of study, but current trends is that they are not and do not need to be separate knowledge bases. Yet, knowledges of students with disabilities and learning needs are integrated into general education, and the advent of dual certification programs has been exceptional at doing this successfully:
  2. Lines 37-48: This is a fundamental misconception of what full inclusionists believe. In fact the reason the knowledge base of inclusion departed from mainstreaming is because there has been a built of knowledges and practices that allow for students with even significant disabilities to be educated in general education settings. To say that there is no specialized training for the workforce to meet the needs of kids with disabilities is incorrect and does not capture the work of those that promote inclusion. Yet, most people who advocate for inclusion would agree that teaching is more complicated than rocket science. The foundational assumptions being made here, yet not clarified, that special education as a field is flawless in who gets labeled, segregated, and so forth is being taken for granted while being argued against in this paper. 

  3. The quote Line 49 is not clearly cited. Is that footnote 11? Also, I take issue with the line in the quote (line 50). The idea is being purported that one's race, ethnicity, gender, etc. have nothing to do with learning. But of course, they cannot be untangled.
  4. The authors argue that those advocating for full inclusion are arguing that no students need to be labeled, singled, out, etc. Although it is true that we now have the knowledge base to integrate general educaiton and special eudcation so that kids with disabilities can be included, most people advocating for inclusion are not claiming that no child should be labeled with a disability category. Certainly caution in some problematic assessment practices is needed (hence our issues wiht overrperenstation), but instead disability is promoted as something one can be proud of, and labeling it is not in itself a problem. Yet, it is true that segregation is not necessary, because segregation is fundamentally harmful and keeps expectations low (and so and so forth- as built up by decades of research). 

  5. Line 100: The moral taint of segregation is attributed to special education for good reasons, it is harmful to children, it keeps thier expectations low, and minority students are far more likely to be segregated. It is a flawed assumption to assume that disability is a neutral concept, or that special education is a benevolent concept. 
  6. I find it interesting that the author(s) cite disability studies scholars to argue against the project of inclusion, but also say that RTI is a project of full inclusionists. In fact, most disability studies scholars would agree with the author(s) on the basic premise that RTI does not align strongly to promoting full inclusion in schools. 
  7. Line 231: Inclusive environments have proven to be less successful than specialized separate: This is not giving the body of literature on the topic any creedance. The body of literature states the opposite, this one citation does not speak for the larger body of literature.
  8. I think the point and questions raised are reasonable questions to raise (two tiers or three?) This one, not that? Stigma associated with higher tiers. But, I'm not following the fundamental goal of the article-- is it to simply to reiterate the authors problems with inclusion? I don't think the argument is being made clearly.

Reviewer 2 Report

This paper was a joy to read and makes an important contribution to the contested field of inclusive and special education. Written in a highly engaging and creative style, the authors present a clearly articulated, balanced and practical argument for 'dedicated' special education, or indeed, the need for specialisation within any profession. It was refreshing to read!

The discussion about two or three tiered interventions is enlightening and the place of special education within any tiered system raises important questions about the often unquestionable empirical efficacy of tiered systems. I think this paper  will prompt much pause for reflection on some long held assumptions about tiered interventions. While the authors recognise that some tiered systems have resulted in greater levels of inclusion in regular schools, empirical validation is limited when it comes to demonstrating the efficacy of tiered systems on improved outcomes for children and young people with learning needs and disabilities. 

The authority with which the authors write is evident throughout. It is decisive, balanced, and underpinned by a broad and diverse range of literature.

I have no suggestions to improve this excellent paper, and believe it makes an outstanding contribution to the special issue. Congratulations to the authors. 

Author Response

Thank you for understanding the arguments and seeing their merit.

Reviewer 3 Report

Drawing on the postulates of Hallahan or Kauffman, among others, the authors show "another face" of inclusive education. In my opinion, their greatest virtue is their attempt to unmask some of the less clear and more controversial scenarios of the inclusive education movement, so unthinkingly assumed by many of today's researchers. 

Author Response

Thank you for your understanding, kind comments, and enourgement.

Round 2

Reviewer 1 Report

Most of the concerns that I pointed out in the first round of reviews, and the authors simple "disagreement" with my feedback does not, in my opinion, change the article enough to warrant publication. Several more examples and references were added, but yet these changes don't alter the flaws that were and that remain present in this version of the paper.

The addition in lines 26-32 gets the purpose of both MTSS and inclusive education wrong. MTSS by definition is a a prereferral process. So the premise of the article that MTSS is pushing that general education will take on special education is flawed at the outset. Although it's true that the ideas of MTSS are to hope that fewer kids who can be supported through general education will have their needs met, and thus not need special education services, it is not the case that MTSS presume that all of the special education system will be met through general education. The author(s) also give no reference for this claim.

Line 39-- When you say some argue no special education is needed, who are those some? There is no reference provided to back up the claim. 

Beyond these comments, the first concerns I laid out remain. I could list 100s of articles here that demonstrate, through various methodologies and lenses, that inclusive education does in fact benefit children. However, I'm doubtful that even in doing so the authors will be persuaded to look at the full research body on the topic.  Considering inclusion is not just something promoted in the US, but is the framework promoted by the United Nations should be one clear example that inclusion is far more beneficial than the one citation (line 250)  https://en.unesco.org/themes/inclusion-in-education 

Author Response

Differences of opinion regarding many basic assumptions about full inclusion or inclusionary education v inclusive and special education exist. These disagreements can prevent publication of any manuscript that addresses the issue. If this reviewer will not be satisfied until all disagreement is resolved, then what the reviewer calls "flaws" cannot be satisfied. Other comments are addressed in revision.

The reviewer's comments RE MTSS (and line 39) are addressed in revision.

RE the reviewer's last comment, the argument has never been that inclusion benefits no children. The argument is that inclusion does not benefit all children. Moreover, U.S. law prohibits having no option for students with disabilities other than inclusion. It is not clear what the United Nations is promoting, and a recent analysis of the CRPD is cited.